# Metabolic Acidosis in Patients with Chronic Kidney Disease: Diagnosis, Pathogenesis, and Treatment—A Narrative Review

**DOI:** 10.3390/diagnostics15162052

**Published:** 2025-08-15

**Authors:** Justyna Korus, Maciej Szymczak, Maciej Gołębiowski, Julia Rydzek, Krzysztof Majcherczyk, Jakub Wilk, Kacper Bułdyś, Sławomir Zmonarski, Tomasz Gołębiowski

**Affiliations:** 1Faculty of Medicine, Wroclaw Medical University, Borowska 213, 50-556 Wroclaw, Poland; maciej.golebiowski@student.umw.edu.pl (M.G.); julia.rydzek@student.umw.edu.pl (J.R.); krzysztof.majcherczyk@student.umw.edu.pl (K.M.); jakub.wilk@student.umw.edu.pl (J.W.); 2Department of Nephrology, Transplantation and Internal Medicine, Wroclaw Medical University, Borowska 213, 50-556 Wroclaw, Poland; maciej.szymczak@umw.edu.pl (M.S.); slawomir.zmonarski@umw.edu.pl (S.Z.); tomasz.golebiowski@umw.edu.pl (T.G.); 3Department and Clinic of Hematology, Cellular Therapies and Internal Medicine, Wroclaw Medical University, Ludwika Pasteura 4, 50-367 Wroclaw, Poland; kacper.buldys@student.umw.edu.pl

**Keywords:** metabolic acidosis, chronic kidney disease, normal anion gap metabolic acidosis, high anion gap metabolic acidosis

## Abstract

Metabolic acidosis is a common complication of chronic kidney disease (CKD). The kidneys play a crucial role in acid–base balance, maintaining pH within the normal range (isohydria) by following mechanisms: bicarbonate reabsorption, ammogenesis, and titratable acidity. The anion gap describes the amount of unmeasured anions and is classically evaluated as the difference between the major cation (sodium) and the sum of the two major anions (chloride and bicarbonate). Metabolic acidosis can be divided into two types: normal anion gap metabolic acidosis and high anion gap metabolic acidosis. A high anion gap level is considered unfavorable in terms of prognosis as it is associated with increased mortality. Treatment of metabolic acidosis in patients with chronic kidney disease, despite available therapeutic options, is a challenge. Supplementation with bicarbonates does not improve prognosis on the one hand, and on the other hand, it may be harmful. The new KDIGO guidelines for 2024 have been significantly modified compared to 2012 after negative results of studies on bicarbonate supplementation. Bicarbonate supplementation is currently recommended only when levels are less than 18 mmol/L. This review provides an overview of the current knowledge on the pathophysiology, classification, and therapeutic options, including dietary recommendations and new pharmacology agents.

## 1. Introduction

The kidneys play a crucial role in maintaining proper acid–base balance, which is why metabolic acidosis (MA) is a common complication in patients with chronic kidney disease (CKD) [1] In the Chronic Renal Insufficiency Cohort (CRIC) study, its prevalence among patients with CKD was estimated at 7% in stage 2 CKD, 13% in stage 3, and 37% of participants in stage 4 [2]. Three parameters play a significant role in the assessment of the degree of advancement and diagnosis: the measurement of bicarbonate concentration, pH, and the anion gap. The criterion defining the diagnosis of metabolic acidosis (MA) is the concentration of bicarbonates (HCO_3_^−^) in the serum lower than 22 mmol/L. Additionally, to assess MA, serum pH measurement is used, and maintaining its values within the normal range, i.e., 7.35–7.45, is strictly regulated by the kidneys and lungs [3]. pH values allow us to determine the depth of acidosis, and additional assessment of pCO_2_ and HCO_3_^−^ provides information about the nature of the disorder and whether the acidosis is compensated or not. In patients with CKD, MA is classically observed with low values of pH, pCO_2_, and HCO_3_^−^. The third element in the diagnosis is the assessment of the anion gap, which allows for further categorization of acidosis—with a normal or high anion gap [4].

The main complications of MA in the course of CKD include bone demineralization [5], proteolysis of muscle proteins [6], insulin resistance [7], hypertension [8], progression of CKD [9], and increased mortality [10]. MA adversely affects the prognosis of patients with CKD by accelerating the progression to end-stage renal disease [11]. In the CRIC study, it was shown that in patients with serum bicarbonate levels < 22 mmol/L, the progression of CKD, defined as a 50% reduction in eGFR or reaching ESRD, occurred significantly more often, with an odds ratio of 0.97 (0.94–0.99) for each 1 mEq/L increase in serum bicarbonate [12]. In the following article, we summarize the current state of knowledge on metabolic acidosis.

## 2. Definition and Epidemiology of Metabolic Acidosis in Chronic Kidney Disease

MA is an acid–base balance disorder characterized by a primary excess of acids or a deficiency of bases (HCO_3_^−^), and a decrease in blood pH < 7.35. In clinical practice, particularly in patients with CKD, MA is diagnosed when the concentration of bicarbonates (HCO_3_^−^) in venous blood falls below 22 mmol/L, and it is a common complication [3].

Importantly, MA is not merely a side effect of CKD but also actively contributes to progression through several mechanisms. Chronic retention of hydrogen ions in the nephron stimulates the expression of profibrotic cytokines such as TGF-β1, promoting interstitial fibrosis. Simultaneously, MA activates the renin–angiotensin–aldosterone system (RAAS), leading to sodium retention, renal vasoconstriction, and increased oxidative stress. These changes accelerate tubulointerstitial inflammation and nephron loss. Moreover, increased ammonia production and complement activation further exacerbate renal injury [3,13].

On a systemic level, MA stimulates skeletal muscle catabolism and bone resorption, raising circulating calcium and phosphate levels, which predisposes patients to vascular calcification and nephrocalcinosis. It also alters endocrine regulation, notably by increasing parathyroid hormone (PTH) and fibroblast growth factor 23 (FGF23) secretion, impairing mineral homeostasis and further burdening kidney function. Persistent acidosis ultimately disrupts proximal tubular adaptive mechanisms, accelerating nephron dropout and progression to end-stage kidney disease [14,15].

The development of MA strongly correlates with a decrease in the glomerular filtration rate (GFR), and its prevalence among patients with CKD is higher in cases of more advanced kidney disease [16]. The NHANES III study showed that MA was observed in 1.3% of patients in stage G3, 2.3% in stage G4, and 19.1% in stage G5 [2].

In patients treated with renal replacement therapy, the occurrence of MA depends on the type of renal replacement therapy. As part of the analysis, the prevalence of MA among patients undergoing peritoneal dialysis and hemodialysis was compared. It was shown that in patients undergoing peritoneal dialysis (PD), it occurred less frequently than in patients undergoing hemodialysis (HD) (3% vs. 39%, *p* < 0.05) [17]. In another study involving 141 patients (100 HD and 41 PD), it was found that 73% of patients treated with HD had MA, while only 12% of patients undergoing PD had it [18]. The lower occurrence of MA in patients treated with PD compared to HD is due to several factors. During PD, there is a high absorption of bicarbonates from the dialysate fluid. This difference may also be attributed to the type of buffer used in dialysate solutions. In PD, bicarbonate-based or mixed-buffer solutions (containing both bicarbonate and lactate) are increasingly utilized, offering more physiological acid–base correction. While lactate-buffered solutions require hepatic conversion to bicarbonate—which may be inefficient in patients with impaired liver function and can potentially lead to inadequate correction of acidosis or metabolic alkalosis—they are also associated with dialysate pH and high lactate content, which have been reported to cause discomfort or abdominal pain during infusion [19]. To address these limitations, biocompatible solutions with bicarbonate or mixed buffers have been developed. These solutions demonstrate better metabolic tolerance, reduce oxidative stress on peritoneal mesothelial cells, and have been shown to preserve residual renal function (RRF) and peritoneal membrane integrity for longer durations [20].

In conclusion, the choice between bicarbonate and lactate as a buffer for peritoneal dialysis often depends on the individual patient’s needs and clinical context. Bicarbonate is generally preferred in many cases because it more closely mimics the body’s natural acid–base balance and helps to correct metabolic acidosis more effectively. It can lead to better overall acid–base status and has fewer side effects compared to lactate.

By raising serum bicarbonate levels, both types of renal replacement therapy treat MA; however, HD is an intermittent treatment, whereas PD is a constant one. As a result, bicarbonate levels in the case of HD can vary based on when the test is performed, with lower amounts seen prior to dialysis and acidosis being rectified following HD [21]. Moreover, in individuals with CKD undergoing PD, a higher level of residual renal function (RRF) is maintained [22]. The observation was first confirmed in a study comparing the decline in residual GFR (residual Glomerular Filtration Rate, rGFR) over a period of 18 months in two groups of patients. In the first group undergoing PD, a smaller decrease in rGFR was observed compared to the HD group over the 18 months since the start of treatment, which, as the authors suggest, was associated with better hemodynamic stability. Additional mechanisms contributing to better RRF values in PD include smaller fluctuations in volume and osmotic pressure, which help maintain hemodynamic stability [23]. This phenomenon is associated with stable capillary pressure in the renal glomeruli and a relatively constant level of glomerular filtration. In contrast to this course, HD is associated with episodes of renal ischemia due to rapid changes in serum osmolality and changes in circulating blood volume [24].

## 3. Diagnosis of Metabolic Acidosis

There are no specific symptoms of MA, and symptoms such as nausea, weakness, and shortness of breath may be associated with other complications of CKD, such as water–electrolyte disturbances or those related to toxemia. Previously, MA was diagnosed when laboratory tests in patients with CKD showed bicarbonate levels in venous blood reaching values < 22 mmol/L.

The recommendations of the Working Group on Metabolic and Endocrine Disorders of the Polish Nephrological Society advise that all patients with CKD undergo a test for bicarbonate concentration in venous plasma or blood to check for the presence of MA. In patients with CKD stage 4 or 5, bicarbonate levels in plasma or venous blood are measured at least once a year [25]. MA in individuals with CKD should be diagnosed when the bicarbonate concentration in serum or venous blood is ≤22 mmol/L (Table 1). It is also important to determine the nature of acidosis, i.e., uncompensated or compensated (Table 2). Due to the risk of bleeding and artery injury from puncturing, it is not advised to collect arterial blood samples. There are no strict guidelines on when to start assessing HCO_3_^−^ and how often to repeat it. Considering the epidemiological data, it seems reasonable to perform the test in G4 or G5 and monitor HCO_3_^−^ during therapy.

Additionally, patients’ levels of serum creatinine, albumin, urea, sodium, potassium, calcium, phosphorus, glucose, and CRP should be tested [26].

Diagnosis of high anion gap metabolic acidosis should be based on the assessment of renal failure and possible substances taken by the patient (Figure 1). In the diagnosis of metabolic acidosis with a normal anion gap (hyperchloremic) (Figure 2), renal tubular acidosis (RTA), both distal (type 1) and proximal (type 2), should be considered. Such a picture may resemble the compensation of respiratory alkalosis; therefore, it is necessary to confirm acidosis by measuring blood pH [27].

Tubular acidosis requires specific diagnostics, in which other parameters are used, including urine pH, sodium concentration in urine, and ammonia in the urine. The importance of these tests is presented in the discussion of these disorders.

## 4. Special Conditions to Consider in a Patient with CKD and MA

### 4.1. The Effect of Albumin Concentration on the Anion Gap

When assessing AG, one should take into account conditions associated with hypoalbuminemia. Albumins are anionic proteins; therefore, a lower concentration leads to a decrease in AG. Hypoalbuminemia (albumin concentration < 3.5 g/dL) is common among patients with CKD and results from decreased synthesis (reduced nutrient intake, inflammatory processes, and acidosis) and increased degradation of albumin in the body (proteinuria, loss of albumin during dialysis) [32,33]. Hypoalbuminemia causes a decrease in AG, which was confirmed in studies by Feldman et al., who determined that each decrease in serum albumin by 10 g/L results in a decrease in AG by 2.3 mmol/L [34]. AG must be corrected when the albumin concentration decreases. This can be completed using the formula: corrected AG = measured AG + 2.5 × (4.0 − albumin [g/dL]) [28]. Hypoalbuminemia can mask an elevated anion gap, associated with high concentrations of the main anions Cl^−^ and HCO_3_^−^ [35].

### 4.2. Chronic Obstructive Pulmonary Disease

Among the group of people suffering from chronic obstructive pulmonary disease (COPD), the main complication is the development of hypercapnia. However, the problem with gas exchange occurring in these patients leads to the retention of carbon dioxide (pCO_2_), which is then hydrated, resulting in the formation of H^+^ ions. The process is illustrated by the reaction: CO_2_ + H_2_O → H_2_CO_3_ → HCO_3_^−^ + H^+^. The consequence is the development of respiratory acidosis. Compensatory mechanisms focus on increased reabsorption of HCO_3_^−^ by the kidneys. A compensatory response to acidosis occurs, involving a secondary increase in bicarbonate levels, while arterial blood gasometry shows a decrease in pH, an increase in pCO_2,_ and HCO_3_^−^. In patients with CKD and coexisting COPD, the degree of compensation for metabolic acidosis associated with decreased pCO_2_ should be assessed. The pCO_2_ concentration under normal conditions is 35–45 mmHg. In the case of metabolic acidosis (MA) in the course of chronic kidney disease (CKD) and a decrease in HCO_3_^−^ concentration, a proportional decrease in pCO_2_ is observed, which should be equal to the value given by the Winters formula: Paco2 = ([1.5 × HCO_3_^−^ + 8] ± 2) [28]. Impaired respiratory response will be associated with a higher pCO_2_ than calculated in this formula. Another method is to add 15 to the current HCO_3_^−^ concentration [36]. These methods allow for the determination of mixed acidosis (metabolic–respiratory).

### 4.3. Lactic Acidosis

Lactic acid (lactate) is produced as a result of the degradation of pyruvate by lactate dehydrogenase in anaerobic metabolism. It is an intermediate product of carbohydrate metabolism produced by skeletal muscles, the brain, or the kidneys [37]. The kidneys participate in removing excess lactate from circulation through the Cori cycle. There are two types of lactic acidosis. Type A is associated with impaired tissue oxidation, resulting from hypovolemia, sepsis, or failure. Type B is less common, and three subtypes are distinguished: lactic acidosis B1 caused by diseases (such as asthma and diabetes), lactic acidosis B2 induced by drugs or toxins (such as cyanide, metformin, or epinephrine), and lactic acidosis B3 caused by hereditary metabolic disorders. Lactic acidosis is one of the most common complications among critically ill patients [38,39]. Accumulated lactate causes a decrease in blood pH and a reduction in bicarbonate concentration, leading to the development of HAGMA, because in lactic acidosis, lactate anions (Lac-) replace HCO_3_^−^, not chlorides, which are the basis for calculating the anion gap [28].

### 4.4. Diabetic Ketoacidosis

The cause is the accumulation of keto acids, such as β-hydroxybutyric acid and acetoacetic acid, in the extracellular space [40]. The increase in their secretion is due to insulin deficiency, which prevents the body from utilizing glucose, so it starts burning fat. Increased concentration of ketoacids lowers the value of HCO_3_^−^ and raises AG. In patients with ketoacidosis, there is often an increase in AG because ketoacids have the properties of anions [41]. Insulin inhibits lipolysis and reduces the influx and oxidation of fatty acids in the liver, which quickly limits the production of ketoacids. Anions of ketoacids, which previously replaced the lost bicarbonate, can lead to its reformation after oxidation. Therefore, the correction of ketoacidosis often occurs alongside insulin treatment [42].

### 4.5. Salicylates

Salicylate poisoning leads to many metabolic disturbances. Initially, there is stimulation of the respiratory center in the medulla oblongata, resulting in accelerated and deepened breathing and the development of respiratory alkalosis [43]. At the same time, salicylates disrupt mitochondrial function by uncoupling oxidative phosphorylation, forcing cells to switch to anaerobic metabolism [28]. As a result, the concentration of lactic acid in the blood increases, which—along with the metabolites of salicylates, such as salicyluric acid and phenolic glucuronide of salicylate [44]—leads to the development of MA. The body initially tries to compensate for the disturbances by increasing hyperventilation. However, over time, the patient’s body is unable to further compensate for respiratory disturbances, leading to an exacerbation of metabolic acidosis. As a consequence, there is a destabilization of circulation and damage to vital organs [45].

### 4.6. Drug-Induced Acidosis

Many medications can contribute to the occurrence of metabolic acidosis. Although this type of acidosis usually progresses mildly, in some cases, it can lead to life-threatening disorders. Its pathophysiology is associated with the following mechanisms: exogenous acid load, increased endogenous acid production, impaired renal acid excretion, and bicarbonate loss. The diagnosis is based on identifying the type of acid–base balance disorders, correlating with drug exposure, and confirming abnormalities in laboratory tests [46].

In the context of chronic kidney disease (CKD), two classes of drugs widely used in this population draw particular clinical attention—metformin and sodium–glucose cotransporter type 2 (SGLT2) inhibitors.

(a)Metformin

It is the primary medication for patients with type 2 diabetes and CKD, but its use is associated with a rare yet potentially fatal risk of lactic acidosis (MALA), especially in clinical situations such as hypoxia, sepsis, circulatory collapse, or kidney or liver failure [47]. The mechanism of toxicity involves the inhibition of mitochondrial complex I and pyruvate carboxylase, leading to the accumulation of lactate. The diagnosis of MALA is based on the presence of metabolic acidosis with a high anion gap and a lactate concentration > 5 mmol/L in a patient taking metformin. This medication can be safely used with eGFR ≥ 45 mL/min/1.73 m^2^ and with eGFR 30–44 mL/min/1.73 m^2^—with caution. Management includes supportive treatment, such as hemodynamic stabilization, acid–base balance control, and, if indicated, elimination of metformin through intermittent hemodialysis, preferred over CRRT. In severe cases, especially with resistant hypotension, the use of VA-ECMO can be considered, often in conjunction with renal replacement therapy [48].

(b)SGLT2 Inhibitors

A group of drugs widely used in nephroprotective and cardiometabolic therapy in patients with CKD can lead to the development of euglycemic ketoacidosis (EKA). This condition is characterized by the presence of severe ketoacidosis despite normal or only slightly elevated blood glucose levels and is associated with a moderate to high clinical risk. The pathophysiology of EDKA includes a relative deficiency of insulin and an increase in the concentration of counter-regulatory hormones, resulting in enhanced lipolysis, ketogenesis, and ketone reabsorption in the kidneys. Triggering factors may include fasting, infections, surgical procedures, pregnancy, or alcohol consumption. Early diagnosis and discontinuation of SGLT2i treatment, replenishment of isotonic fluids, and early administration of insulin are crucial for managing acidosis [49,50]. Additionally, despite the absence of severe hyperglycemia, it may be necessary to administer glucose (e.g., 5% dextrose) to continue insulin therapy. Alkalizing treatment, although usually not necessary, may be considered in cases of pH < 7.1 or delayed bicarbonate regeneration, especially in patients with CKD. According to current guidelines, SGLT2i can be safely used in patients with eGFR ≥ 20 mL/min/1.73 m^2^, but close monitoring of metabolic parameters—including pH, HCO_3_^−^ concentration, and ketone bodies—is recommended, especially in situations of increased metabolic stress [51].

### 4.7. Renal Transplant Recipients

Metabolic acidosis is a common disorder observed in kidney transplant recipients, both in the early and late periods after transplantation. It may result from impaired renal tubular function, chronic graft damage, comorbid conditions (e.g., diabetes), as well as the effects of immunosuppressive drugs, especially calcineurin inhibitors such as tacrolimus [52]. One of the main forms of acidosis in this group of patients is hyperchloremic metabolic acidosis, often manifesting as renal tubular acidosis (RTA), most commonly of the distal type (type 1). Tacrolimus can induce dysfunction of distal tubular cells, disrupting the secretion of hydrogen ions by reducing the activity of the H^+^-ATPase pump and weakening the expression of carbonic anhydrase. This results in impaired urine acidification and compensatory hyperchloremia. Characteristic features include alkaline urine (pH > 5.5), hypokalemia, and persistent hyperchloremic metabolic acidosis. Treatment includes sodium bicarbonate supplementation, correction of potassium deficiency, and consideration of immunosuppression modification—either by reducing the dose of tacrolimus or switching it to another drug (e.g., cyclosporine) if symptoms are severe or persist chronically [53].

## 5. Acid–Base Regulation of the Kidneys

The physiological concentration of hydrogen ions (H^+^) in blood plasma is 35–45 nmol/L, which keeps the pH of arterial blood within the range of 7.35–7.45. The correct concentration of H^+^ ions in plasma is maintained by the kidneys, lungs, liver, and digestive tract.

The kidneys are responsible for maintaining a proper pH level through two different mechanisms [12,27].

The first process involves the reabsorption of filtered bicarbonate. The proximal tubules (PT) absorb 85–90% of it; the Henle loop absorbs 10%, and the collecting ducts absorb the remaining 5–10% [54]. Various types of carbonic anhydrase (CA)—membrane-bound and cytoplasmic—are involved in this process [10]. In the PT, the filtered bicarbonate reacts with H^+^ ions to form carbonic acid (H_2_CO_3_), which is broken down into CO_2_ and H_2_O by the enzyme carbonic anhydrase IV (CA IV). Then CO_2_ diffuses into the tubular cells, where it reacts with H_2_O in the presence of carbonic anhydrase II (CA II) to recombine with HCO_3_^−^ and H^+^. HCO_3_^−^ is transported to the blood, and H^+^ ions return to the PT [54] (Figure 3).

It should be mentioned that some substances, including glucocorticoids, angiotensin II, endothelin, hypokalemia, hypovolemia, or parathyroid hormones, promote the reabsorption of bicarbonates and alter the degree of MA [27].

The second process involves the production of bicarbonate ions as a result of ammoniagenesis and the excretion of titrable acids.

The precursor for ammonia (NH_3_) synthesis in the kidneys is glutamine. In the cells of the proximal tubules (PTs), glutamine is converted to α-ketoglutarate under the influence of the enzyme glutaminase, resulting in the formation of two molecules of NH_3_ and two bicarbonate ions (HCO_3_^−^). Ammonia diffuses into the tubular lumen, where it combines with H^+^ ions to form ammonium ion (NH_4_^+^), which does not easily cross cell membranes and is excreted in urine, connected with chloridium [55]. At this time, the bicarbonate ions formed are transported to the general circulation, contributing to the regeneration of base reserves and maintaining acid–base balance [56].

In the regulation of metabolic acidosis, the excretion of so-called titratable acids, which include phosphoric acid (H_3_PO_4_) and sulfuric acid (H_2_SO_4_). These compounds are removed from the body mainly through glomerular filtration and tubular secretion. The most important buffer enabling the excretion of hydrogen ions in this form is the hydrogen phosphate ion (HPO_4_^2−^), which, by binding H^+^ in the tubular lumen, forms the dihydrogen phosphate ion (H_2_PO_4_^−^). In this form, phosphate is excreted in urine, contributing to the removal of excess hydrogen ions from the body [3,12].

## 6. Metabolic Acidosis and Progression of Chronic Kidney Disease

MA accompanying CKD develops as a result of an imbalance between the amount of acids produced in the body and the impaired ability of the kidneys to eliminate them and produce HCO_3_^−^ ions that serve as a buffer.

A number of factors have been identified that may play a role in the interaction of MA with the progression of CKD. These include complement activation triggered by ammonia and increased production of aldosterone and endothelin-1 (ET-1).

### 6.1. Ammonia

Due to interstitial fibrosis and renal tubule contraction, a common occurrence of CKD is the progressive loss of functional nephrons. The PT tubular cells’ absorption of glutamine is reduced in these circumstances, which lowers the amount of NH_3_ produced and secreted in the collecting ducts [57]. In order to maintain homeostasis, the remaining nephrons produce more NH_3_ to compensate for the loss of acid removal function. However, such a response is harmful to the remaining nephrons [58]. At the molecular level, renal ammoniagenesis primarily occurs in the proximal tubule and involves key enzymes and transporters responsible for the synthesis and handling of ammonia. Glutaminase 1 (GLS1), localized to the mitochondria of PT cells, catalyzes the conversion of glutamine to glutamate and ammonia, initiating the process [59]. Phosphoenolpyruvate carboxykinase (PEPCK), a rate-limiting enzyme in renal gluconeogenesis and ammoniagenesis, facilitates the utilization of glutamate-derived carbon skeletons [60]. Additionally, sodium–hydrogen exchanger 3 (NHE3), located on the apical membrane of proximal tubular cells, plays a critical role in proton secretion and indirectly supports ammonia trapping in the tubular lumen [61]. In CKD, the dysregulation of these enzymes and transporters contributes to impaired ammonia generation and altered acid–base compensation mechanisms, while at the same time promoting intrarenal toxicity via retained or mislocalized ammonia.

Researchers Goraya N. et al. proposed a hypothesis that high levels of NH_3_ activate the alternative complement pathway through amidation. Ammonia can combine with C3 protein and activate the alternative complement pathway since it is a nitrogen nucleophile. Damage to the tubules and interstitium is linked to elevated ammonia levels in the kidneys, and this activation is the exact cause of its detrimental consequences. This promotes the development of tubulointerstitial nephritis by causing the production of complement components that have cytotoxic and inflammatory effects [62]. The result of this process is the induction of chemotaxis, which is a mechanism allowing cells, mainly of the immune system, to move in a directed manner towards an area with a higher concentration of chemotactic molecules [63]. Then, a membrane attack complex C5b-9 is formed, and phagocytosis is stimulated, leading to the ultimate damage of renal tubular cells [10]. In one study, NaHCO_3_ supplementation decreased complement component deposition and tubular-interstitial damage in rats following nephrectomy when compared to the control group. Bicarbonate use decreases the complement system’s activation surrounding the renal tubules, which is linked to environmental alkalization, ammoniagenesis inhibition, and decreased ammonia synthesis in the renal cortex [64].

Collectively, these observations highlight that impaired regulation of key ammoniagenic pathways—particularly involving GLS1, PEPCK, and NHE3—not only reduces the kidney’s capacity for acid excretion but also contributes to maladaptive increases in ammonia that promote inflammation, complement activation, and further nephron injury. Understanding these molecular mechanisms provides a biologically plausible link between acid–base disturbances and CKD progression and may inform future therapeutic strategies aimed at preserving nephron integrity by modulating renal ammoniagenesis [61].

### 6.2. Aldosterone

Another factor influencing the progression of CKD in the course of MA is aldosterone. Its production begins in the kidneys, where juxtaglomerular cells release renin, which converts angiotensinogen into angiotensin I. Then, the angiotensin-converting enzyme (ACE) leads to the conversion of angiotensin I into angiotensin II, which stimulates the adrenal cortex to release aldosterone. Aldosterone, in turn, affects the kidneys, increasing sodium reabsorption and potassium excretion. One of the factors that increases aldosterone synthesis is a higher concentration of potassium in the serum [65]. In CKD, aldosterone secretion is mainly stimulated by angiotensin II and hyperkalemia. Aldosterone, through its action, can cause a decrease in GFR levels because it acts profibrotically [66]. Mineralocorticoid raises blood pressure and pressure in the renal glomeruli through AT_1_ receptors. They cause vasoconstriction, including the efferent arterioles in the glomerulus, which leads to an increase in glomerular pressure and elevated blood pressure. Additionally, angiotensin II acts pro-inflammatory and profibrotic by stimulating the production of cytokines (e.g., TGF-β), which promote fibrosis of the renal parenchyma, thereby causing progressive kidney damage [67,68].

### 6.3. Endothelin

It has been shown that acidosis also increases tubular-interstitial damage caused by endothelin (ET). It is a peptide derived from endothelial cells, with three isoforms: ET-1, ET-2, and ET-3 [69]. The most significant for kidney function is ET-1, whose action through connective tissue growth factor and transforming growth factor β1 (TGF-β1) promotes inflammatory processes and stimulates kidney fibrosis [3]. In the course of MA, there is an increased production of ET-1 in the kidneys, which consequently stimulates the secretion of H^+^ ions in the proximal and distal segments of the nephron and reduces the secretion of HCO_3_^−^ [69]. In a study conducted on rats, one group received a solution of (NH_4_)_2_SO_4_, while the other received water. In rats receiving (NH_4_)_2_SO_4_, increased ET-1 secretion in the kidneys was observed. The administration of bosentan—a drug-blocking ET-1 receptors—resulted in increased secretion of HCO_3_^−^ ions and weakened the ability of distal tubules to acidify urine [70]. In another study on rats, it was shown that a high-protein diet accelerates the progression of CKD, worsens GFR, and causes tubular-interstitial damage, and this process is dependent on ET activity. The introduction of alkalizing therapy effectively prevented these negative effects [10].

## 7. Consequences of Metabolic Acidosis in Chronic Kidney Disease

MA in patients with CKD disrupts numerous metabolic pathways, resulting in the improper functioning of many organs and systems in the body [12]. One of these mechanisms is the development of the malnutrition–inflammation–atherosclerosis (MIA) syndrome associated with protein-energy malnutrition or an inflammatory state [71].

There are data indicating that an acidic environment with low pH exacerbates inflammation by affecting macrophage functions. In one study, it has been shown that in an acidic environment, macrophages exposed to bacterial lipopolysaccharides release more cytokines, such as tumor necrosis factor (TNFα) or interferon-γ (IFN-γ) leads to increased expression of the inducible isoform of nitric oxide synthase (iNOS or NOS II), which is responsible for the increased production of nitric oxide (NO) [72,73]. The increase in NO levels mediates the process of vasodilation [74]. In studies conducted on rats, acute metabolic acidosis was associated with a more pronounced decrease in mean arterial pressure than in the group of animals with normal pH [75]. Additionally, as a result, there was an increase in the levels of NO metabolites in the plasma, indicating higher NO production in acidic conditions [76]. Generally, MA leads to impaired contractility of the heart muscle, decreased blood pressure, and reduced blood flow through the liver and kidneys [77].

One of the extrarenal consequences of MA is its unfavorable impact on calcium-phosphate metabolism. One of them is bone demineralization. It occurs as a result of increased osteoclast activity and inhibition of osteoblast function, as well as a negative calcium balance [3]. MA leads to decreased sensitivity of calcium receptors through the stimulation of parathyroid glands to secrete parathyroid hormone [78], the release of mineral phosphorus compounds and HCO_3_^−^ from bones [79], and an increase in the severity of bone changes [80]. Observational clinical studies have shown that MA stimulates the proteolysis of skeletal muscles through inflammatory pathways [13,81,82].

MA in CKD negatively affects the cardiovascular system. By acting on ion channels in the heart, it contributes to the inhibition of sodium conduction. The result of this process is the prolongation of the QT interval in the ECG, which can be corrected by administering bicarbonates to the patients. Moreover, studies confirm the impact of MA on the development of arrhythmias. They result from changes in intracellular pH and membrane potential, the decrease of which, caused by acidosis, shifts the excitability threshold, leading to the development of additional excitations in the cardiac muscle. MA alters the conductivity of the sodium channel by activating the Na^+^/H^+^ exchanger, which removes excess H^+^ ions from the cells in exchange for Na^+^ ions. In the long term, this leads to the activation of the Na^+^/Ca^2+^ exchanger and the influx of Ca^2+^ ions into the cells. Arrhythmias induced by acidosis include early afterdepolarizations, triggered activity induced by acidosis, delayed afterdepolarizations, and reentry [83]. Affecting gap junctions, to which connexin belongs, it worsens conductivity in the cardiac muscle, thereby predisposing to blocks [3]. Its detrimental impact on the cardiovascular system was examined in research looking at the connection between MA and pulse wave velocity. It has been demonstrated that patients with MA and CKD have greater arterial stiffness. This has to do with many processes. Above all, MA promotes the release of calcium and phosphates from the bones into the blood by causing bone resorption [30]. In CKD, increased concentrations of pro-inflammatory cytokines are observed. Interleukin-1 is also attributed with proatherogenic activity. By modulating the activity of ion channels (Ca^2+^, K^+^), it leads to an extended repolarization time and additionally affects the inflammatory pathways in the cardiac muscle, resulting in mitochondrial damage [84]. In conducted observational studies, it has been shown that the presence of MA correlates with increased overall mortality [85], which was associated with the following diseases: myocardial infarction (MI), ischemic stroke, incidental heart failure, and hospitalization due to heart failure [86,87].

Chronic metabolic acidosis contributes to urolithiasis through multiple mechanisms. It promotes hypocitraturia, reduces the urine’s buffering capacity, and lowers citrate availability, which normally inhibits calcium salt crystallization. Additionally, low urinary pH facilitates uric acid and calcium oxalate precipitation, while chronic acidosis increases bone resorption, leading to hypercalciuria. In distal renal tubular acidosis (type 1), impaired distal H^+^ secretion results in inadequate urine acidification, favoring the formation of calcium, urate, and cystine stones [31]. Patients with distal RTA are at higher risk of recurrent nephrolithiasis and nephrocalcinosis, potentially accelerating the progression of kidney dysfunction. Early diagnosis and implementation of alkalinizing and citrate-supplementing therapies are essential to mitigate stone formation and preserve renal function [88].

## 8. Anion Gap

The term anion gap (AG) was first described in 1936 by James Gamble, but it was popularized in the 1970s by American researchers James P. Peters and Gerald M. Severing [89].

The normal AG values range from 8 to 12 mEq/L, and its value depends on three electrolytes present in the serum: sodium [Na^+^], chloride [Cl^−^], and bicarbonate [HCO_3_^−^]. To calculate it, the formula is used: AG = Na^−^ − (Cl^−^ + HCO_3_^−^). The AG concentration also includes measurable anions such as albumin or phosphates, as well as unmeasurable ones that accumulate in the course of kidney diseases, such as indoxyl and p-cresol. In the course of CKD, changes in the concentrations of measurable cations (potassium, magnesium, calcium) can also affect the level of AG [90]. Depending on the AG calculation results, we can distinguish between normal anion gap metabolic acidosis (NAGMA) and high anion gap metabolic acidosis (HAGMA).

## 9. Normal Anion Gap Metabolic Acidosis (NAGMA)

NAGMA, also known as hyperchloremic acidosis, is a type of metabolic acidosis (MA) in which the anion gap is maintained within the normal range. As CKD progresses to more advanced stages, the concentration of HCO_3_^−^ in the blood decreases due to impaired reabsorption, while the deficiency of anions is compensated by an increase in the secretion of chlorides and their concentration in the serum [45]. This type is observed in the earlier stages of CKD [91]. As previously described, this process is mediated by the Cl^−^/HCO_3_^−^ cotransporter, which exchanges anions electrostatically neutral, allowing AG to be maintained within the normal range.

### 9.1. Renal Tubular Acidosis

The mechanisms leading to NAGMA in CKD show significant overlap with those observed in renal tubular acidosis (RTA), a group of disorders characterized by impaired acid–base handling at the tubular level. In particular, abnormalities seen in CKD mimic defects found in type 2 (proximal) and type 1 or 4 (distal) RTA. All these tubular kidney disorders contribute to the development of NAGMA [92].

Proximal (type 2) RTA rarely occurs as an isolated defect and is most often accompanied by impaired absorption of other substances, such as phosphates, glucose, uric acid, and amino acids. In adults, proximal RTA most often results from damage to the proximal tubules caused by the presence of monoclonal light chains of immunoglobulins, excreted during the course of monoclonal gammopathies. The development of this type of RTA can also be caused by nephrotoxic drugs, such as tenofovir and ifosfamide (Table 3), as well as autoimmune diseases, e.g., Sjögren’s syndrome and Fanconi syndrome. In children, proximal RTA most commonly occurs in the course of cystinosis, after ifosfamide therapy, or is idiopathic in nature. In treatment, bicarbonate and citrate supplementation, as well as treatment of the underlying disease, are used [93,94].

At the core of type 1 RTA lies a primary defect in impaired acidification in the distal segment of the renal tubule. In adults, the main causes of this disorder are autoimmune diseases, such as Sjögren’s syndrome and rheumatoid arthritis. Another risk factor is the occurrence of hypercalciuria. The hereditary form of RTA is found in children and is associated with genetic mutations in the basolateral chloride–bicarbonate exchanger (*SLC4A1* gene) and the apical hydrogen ATPase (*ATP6V0A4* and *ATP6V1B1* genes). Medications such as ifosfamide or ibuprofen can cause the development of type 1 RTA [93,95].

Some degree of hypokalemia is typically present in type 1 distal and type 2 proximal renal tubular acidosis.

In type 4, the function of the distal and collecting tubules is impaired. There is impaired secretion of aldosterone or resistance of the distal tubule to its action. As a result of these processes, the excretion of H^+^ and K^+^ ions does not occur. Along with this type of acidosis, hyperkalemia occurs. The prognosis depends on the cause. It coexists with diabetes, diabetic nephropathy, or lupus nephropathy. The treatment involves correcting hyperkalemia, using loop or thiazide diuretics, and correcting acidosis with sodium bicarbonate [96].

In the differential diagnosis, other causes of bicarbonate loss should also be considered, such as diarrhea, bladder disorders, recovery from ketoacidosis, or toluene poisoning [97].

### 9.2. Diagnostics of RTA

In the case of unexplained hyperkalemic metabolic acidosis with a normal AG, the possibility of RTA—both distal and proximal—should be considered. Since a similar electrolyte pattern may be the result of compensation for respiratory alkalosis, the first diagnostic step should be to confirm the presence of MA by measuring blood pH [98,99].

After confirming that the AG remains within normal limits, it is worth considering other causes beyond RTA—such as the loss of bicarbonates from the gastrointestinal tract (e.g., due to diarrhea, fistulas, or bladder reconstruction surgeries), the metabolism of anions such as butyrate, acetate, or lactate, which can be converted into HCO_3_^−^, and toluene poisoning (e.g., through inhaling glues or paints) [100].

The next step in the diagnosis should be the determination of urine pH and the assessment of ammonia excretion, which allows for the differentiation of the causes of hyperkalemic metabolic acidosis [99].

#### 9.2.1. Urine pH

Urine pH is an important indicator in differentiating types of renal tubular acidosis. In patients with normal kidney function and normal renal acidification mechanisms, the development of MA usually results in a decrease in urine pH to ≤5.3. Exceptions may include cases of chronic acidosis with hypokalemia (e.g., in the course of diarrhea), where ionic shifts cause an increase in ammonia production and a paradoxical rise in urine pH above 5.5, which can mimic distal RTA [101,102]. In such patients, intracellular acidosis is observed, resulting from the influx of hydrogen and sodium into the cells and potassium out. This induces hydrogen secretion in the kidneys and increased ammonia production, which is excreted into the tubular lumen and combines with hydrogen ions to form ammonium (NH_4_^+^).

In classic distal RTA, urine pH remains ≥ 5.5 despite the presence of acidosis, which is due to impaired H^+^ secretion in the distal nephron.

In proximal RTA, urine pH is variable. It may be elevated after the administration of bases when plasma HCO_3_^−^ exceeds the reabsorption capacity in the proximal tubule. However, with low serum HCO_3_^−^ concentration, the distal nephron functions properly and lowers urine pH to ≤5.3 [103].

In patients with MA and elevated urine pH, other causes than RTA should be considered, such as urinary tract infections with urease-splitting bacteria, chronic hypokalemia increasing ammonia excretion, or decreased intravascular volume limiting sodium delivery to the tubules. Therefore, the interpretation of high urine pH should consider a urinalysis, a possible culture, and a urine sodium concentration above 25 mEq/L [104].

#### 9.2.2. Urinary Excretion of Ammonium

The excretion of ammonia in urine helps differentiate the causes of MA with NAGMA. In distal RTA, urine pH is usually ≥5.5, and ammonia excretion is reduced due to impaired acidification in the distal nephron. However, elevated urine pH can also occur with normal ammonium excretion, for example, in diarrhea or toluene poisoning, where acidification mechanisms are preserved [105].

For this reason, estimating ammoniogenesis may be diagnostically useful. Although the direct measurement of ammonia in urine is the most reliable, it is not widely available. In clinical practice, indirect indicators such as the urine anion gap (UAG) and urine osmolal gap (UOG) are used, although their reliability is limited [106]. The urine anion gap is used to help assess the cause of metabolic acidosis, particularly to differentiate between renal and gastrointestinal causes of non-anion gap acidosis. Following formula is used: UAG = Na^+^ + K^+^ − Cl^−^, where Na^+^ = urine sodium concentration (mEq/L), K^+^ = urine potassium concentration (mEq/L), Cl^−^ = urine chloride concentration (mEq/L). A positive UAG indicates decreased ammonium excretion (distal renal tubular acidosis), while a negative UAG may suggest appropriate ammonium excretion due to gastrointestinal losses of bicarbonate (e.g., diarrhea-induced acidosis). UOG indicates how many dissolved substances are present in a given amount of urine (per kilogram of water). It shows how effectively the kidneys can either concentrate or dilute urine in order to regulate the body’s water balance [107]. The urine osmolal gap helps estimate the amount of unmeasured osmoles in urine, primarily ammonium. The following formula is used: UOG = Measured Urine Osmolality − Calculated Urine Osmolality Calculated Urine Osmolality = 2 × Na^+^ + Glucose/18 + Urea/2.8, where measured urine osmolality is obtained from laboratory analysis, sodium, glucose, and urea concentrations are in mg/dL. A high uric osmolal gap suggests increased ammonium excretion.

In proximal RTA, the measurement of ammonia has no diagnostic value—patients excrete normal amounts of ammonia both before and after treatment, even though urine pH may be abnormally elevated after the administration of bases [105].

#### 9.2.3. Differences in the Diagnosis Between Distal Renal Tubular Acidosis and Proximal Renal Tubular Acidosis

In distal RTA, the kidneys are unable to properly excrete hydrogen ions, leading to their accumulation in the body if alkalizing treatment is not applied. Nevertheless, the bicarbonate level in serum usually does not drop below 10 mEq/L. For the diagnosis of distal RTA, a basic urine pH (≥5.5), urine sodium concentration exceeding 25 mEq/L, and reduced ammonia excretion are typical. Ammonium excretion can be indirectly assessed by calculating the urinary anion or osmolar gap, with the latter method being more accurate, especially in clinical situations that disrupt the reliability of the AG [100,108].

Proximal RTA results from impaired bicarbonate reabsorption in the proximal tubule, leading to their excessive loss in urine, especially when serum HCO_3_^−^ concentration exceeds the reduced reabsorption threshold. Although the distal segments of the nephron can partially compensate for this defect, they are unable to fully take over the functions of the proximal segment. In untreated patients, the bicarbonate level in the blood typically ranges from 12 to 20 mEq/L, and the urine pH depends on the current HCO_3_^−^ concentration [109]. After intravenous administration of sodium bicarbonate, urine pH can rise sharply (even above 7.5), and fractional excretion of HCO_3_^−^ (FEHCO_3_) exceeds 15–20% [97,110]. To calculate it, the formula is used:$${FEHCO}_{3}=\frac{{UHCO}_{3}\times SCr}{{SHCO}_{3}\times UCr}\;\times 100$$

Abbreviations: FEHCO_3_, fractional excretion of HCO_3_^−^; SCr, concentration of creatinine in serum (mg/dL or μmol/L); SHCO_3_, concentration of bicarbonate in serum (mmol/L); concentration of creatinine in urine (mg/dL or μmol/L); UHCO_3_, concentration of bicarbonate in urine (mmol/L).

This type of RTA can occur as an isolated defect or be part of Fanconi syndrome, which is characterized by, among others, glucosuria, hypophosphatemia, and aminoaciduria [97].

## 10. High Anion Gap Metabolic Acidosis (HAGMA)

The basis for diagnosing HAGMA is a decrease in pH, a decrease in HCO_3_, and an increase in AG > 12 mmol/L. The most common factors causing it are kidney failure, lactic acidosis, and diabetic ketoacidosis [29]. HAGMA occurs in the later stages of CKD—from stage G4 [111], which is associated with the accumulation of non-chloride anions: sulfate, phosphate, and organic acids [91]. Some studies have shown that in kidney failure, anions such as sulfonic compounds (indoxyl, p-cresol) accumulate, potentially increasing AG [112]. Additionally, the results of the conducted studies provided information that HAGMA is associated with a 3.04-fold increased risk of kidney failure in the case of replacement therapy [111] and a 5.56-fold higher all-cause mortality rate compared to a normal anion gap.

There are also other causes of HAGMA, including those caused by glycols, 5-oxoproline, L-lactic acid, D-lactic acid, methanol, or aspirin (Table 4). It is characterized by a high AG and a high concentration of chlorides. In experiments on animal models, it has been shown that the tubules reabsorb a reduced amount of HCO_3_^−^, accompanied by increased secretion of Cl^−^ into the serum [90].

## 11. Basic Dietary Recommendations for Patients with Metabolic Acidosis

### 11.1. General Information

Diet is one of the methods for correcting MA. Proper nutrition can effectively influence metabolic disorders and slow the progression of kidney disease.

Most Europeans follow a “Western diet”, which is characterized by the consumption of high-protein foods (from fatty and processed meats), refined grains, saturated fats, alcohol, sugars, and salt, with a reduced intake of fruits and vegetables [113]. Particular attention should be paid to the amount of animal proteins consumed, which, due to their role in the production of H^+^ ions, contribute to the development of MA and should be significantly restricted in patients, depending on the stage of CKD. Patients with CKD stages G3–G5 should consume a diet providing 0.55–0.60 g of dietary protein per kg of body weight per day, or a very low-protein diet providing 0.28–0.43 g of dietary protein/kg of body weight/day with additional ketone body/amino acid analogs to meet the protein requirement (0.55–0.60 g/kg of body weight/day) [114,115]. The kidneys in the course of MA are characterized by a reduced ability to excrete H^+^ ions, which, due to accumulation, decrease the body’s pH level, exacerbating the state of acidosis [116,117].

In patients with CKD in the pre-dialysis period, a low-protein diet is recommended; however, it is necessary to supplement with essential amino acids and branched-chain amino acids (BCAA) [118]. An example is a preparation that contains essential amino acids and their α-keto- and α-hydroxy analogs (in the form of calcium salts) [119]. Studies have shown that a diet with reduced protein intake (in the range of 0.6–0.8 g/kg body weight daily) slows the deterioration of kidney function, reduces the risk of MA development, and delays the need to start renal replacement therapy. Thanks to this, it is possible to delay the start of dialysis treatment or to use a smaller number of dialysis sessions per week, known as incremental dialysis, gradually implementing dialysis therapy, which makes this diet an effective form of treatment for CKD and its associated disorders [120].

### 11.2. Low-Protein Diet

Studies show that low-protein diets in patients with CKD slow down metabolic bone disease and secondary hyperparathyroidism. They improve calcium-phosphate metabolism by lowering the level of fibroblast growth factor 23 (FGF23) [121] and help maintain proper bone density by reducing acid load [117,122]. As kidney function deteriorates, the amount of acids excreted by them becomes less than the endogenous production. MA is a factor leading to a decrease in the level of 1,25-dihydroxyvitamin D3 (calcitriol) in serum and an increase in parathyroid hormone (secondary hyperparathyroidism) and FGF23. As kidney function deteriorates, the level of circulating PTH increases, leading to secondary hyperparathyroidism. Elevated PTH levels stimulate osteoblasts, which activate osteoclasts, resulting in the release of calcium and phosphates from the bones. MA additionally increases PTH secretion, weakens collagen synthesis by osteoblasts, and enhances osteoclast activity. This leads to an imbalance between bone formation and resorption, promoting bone mass loss [123,124].

### 11.3. Fruit and Vegetable Diet

A diet rich in fruits and vegetables is the basis of an alkalizing diet and may be advised to patients with MA. It influences the increase in the body’s pH by neutralizing acids through its alkaline action and the presence of organic acids such as citrate and malate [125], which are subsequently metabolized into bicarbonates, among other things [126]. They are responsible for maintaining the proper acid–base homeostasis of the body, improving the net endogenous acid production (NEAP) values, which is the difference between protons absorbed through the metabolism of highly acidic products consumed in the diet, such as animal proteins, and alkalis absorbed from foods mostly composed of basic components, like fruits, further reduced by the anions of organic acids eliminated with urine [16]. A higher NEAP is associated with a lower concentration of bicarbonates, which correlates with the development of MA, so it is important to maintain it in a lower range [12]. Fruits and vegetables also contain potassium, magnesium, and calcium, which are essential in many processes and transformations occurring in the body, ensuring proper homeostasis [127]. A drawback of a diet based on a large amount of fruits and vegetables is the possibility of hyperkalemia in patients, the risk of which increases with a decrease in GFR and occurs in 31% of patients with an estimated glomerular filtration rate (eGFR) ≤ 20 mL/min/1.73 m^2^. To avoid the development of hyperkalemia, patients are advised to limit their potassium intake to 2000–3000 mg/day [128]. Caution should be exercised in individuals with eGFR < 30 mL/min/1.73 m^2^, for whom potassium intake may need to be restricted. Plant-based diets, despite their relatively higher potassium content, have not been shown to cause hyperkalemia in these patients [129]. Blanching and cooking food products allow for the reduction in potassium content in vegetables and legumes, which may decrease the risk of hyperkalemia [128]. For several years, oral potassium-binding resins have also been available, such as patiromer, which is registered for the treatment of hyperkalemia in patients with CKD.

A diet rich in fruits and vegetables can help avoid the adverse consequences of acid–base imbalance, such as osteoporosis, due to excessive use of bone mass. In studies conducted on 3089 people in China, it was shown that higher consumption of fruits and vegetables had a dose-dependent association with greater BMD and a lower risk of osteoporosis [130,131]. By preventing the progression of MA, it also protects muscle tissue from degradation, forming one of the foundations of musculoskeletal prevention in individuals struggling with CKD [132].

This diet proved to be as effective as sodium bicarbonate supplementation in treating MA. This was confirmed by the analysis of a 5-year clinical study conducted on patients with stage 3 CKD and advanced MA, with accompanying macroalbuminuria and without diabetes, who were given vegetables and fruits such as apples, apricots, oranges, peaches, pears, raisins, and strawberries. Patients consumed vegetables and fruits such as carrots, cauliflower, eggplant, lettuce, potatoes, spinach, tomatoes, and zucchini. Patients did not receive any specific dietary instructions and could incorporate the provided products into their diets according to their preferences. Additionally, the study revealed other health benefits, including improved cardiovascular condition (a decrease in systolic blood pressure to <130 mm Hg) and a reduction in unfavorable lipid profile parameters [62].

The KDIGO guidelines, updated in 2024, also emphasize the importance of the above recommendations. The recommendations contained in them suggest avoiding a high protein intake exceeding 1.3 g/kg of body weight. In individuals with CKD and a risk of progression, particularly those in stages G3–G5, it is recommended to maintain protein intake at a level of 0.6–0.8 g/kg of ideal body weight per day, but under supervision, in patients without metabolic burdens, with the necessity of supplementing essential amino acids and ketone analogs [114].

A vegetarian diet is beneficial for patients with MA because it is based on plant proteins, which produce fewer acids and place less strain on the kidneys than animal proteins. Thanks to this, it improves clinical parameters and prognosis in people with CKD and MA [133,134]. Additionally, a diet rich in plant-based products often contains a high amount of fiber, the increased intake of which can reduce the levels of uremic toxins bound to proteins, such as indoxyl sulfate and p-cresyl sulfate, produced during the breakdown of aromatic amino acids by the gut microbiome. In a conducted study involving a group of 40 patients undergoing HD, increased fiber intake over 6 weeks led to a 29% reduction in the free concentration of indoxyl sulfate in the plasma [135].

### 11.4. Fiber-Rich Diet

Patients with CKD are advised to follow a fiber-rich diet, with an intake of 25–30 g per day. However, it should be noted that excessive fiber intake from vegetables and fruits in advanced stages of CKD may increase the risk of hyperkalemia [136]. In a conducted study among patients with CKD undergoing HD, hyperkalemia was detected in approximately 12.5% [137]. Additionally, it is important to remember the appropriate supplementation of vitamins and exogenous amino acids, such as lysine or methionine, for individuals opting for this diet. This can be aided by a commonly used clinical practice preparation containing a set of ketoanalogs of amino acids, preventing their deficiencies, which could have negative effects on the body’s functioning [129]. Similar to the vegetarian diet, the Nordic diet, characterized by the consumption of fiber-rich products, vegetables, and fruits, also exhibits a beneficial alkalizing potential, essential for counteracting MA processes [138].

### 11.5. Diet Rich in Sugars

A diet rich in simple sugars is not suitable in the case of metabolic acidosis. An excessive amount of sucrose can disrupt the gut microbiota and bile metabolism, which promotes the development of acidosis. Consumed sucrose in the gastrointestinal tract is broken down into glucose and fructose. Then, fructose is metabolized mainly in the liver, where it is converted into fructose-1-phosphate, which is a component leading to the formation of lactic acid. This particularly applies to patients with metabolic disorders, such as type II diabetes, which often coexist with acidosis [139]. In combination with a high intake of animal proteins, it can lead to the development or worsening of MA [140].

### 11.6. Ketogenic Diet

The ketogenic diet also deserves attention. Its harmfulness and the necessity to stop or avoid it in patients with MA should be noted. Excessive production of ketones, occurring with the use of this diet, with inadequate supply of products containing phosphorus, magnesium, or potassium, as well as providing an appropriate amount of fats, can also lead to metabolic disturbances in the body, resulting in MA due to increased AG [141]. Therefore, appropriate education for patients about the harmfulness of certain products and diets is necessary, due to their highly acidifying properties, as well as monitoring their metabolism to avoid possible complications of progressive MA, preventing its exacerbations.

## 12. Pharmacological Treatment of Metabolic Acidosis

In the KDIGO guidelines published in 2024, the approach to treating MA in CKD with oral sodium bicarbonate (NaHCO_3_) has been changed. It is no longer recommended for use at HCO_3_^−^ concentrations < 22 mmol/L, and its inclusion is only considered at levels < 18 mmol/L [114]. The authors justify the cautious approach by the lack of unequivocal results from large randomized clinical trials and emphasize the need to conduct them to precisely determine the threshold concentration of HCO_3_^−^ at which the intervention will have a beneficial impact on patient health.

Pharmacological treatment of MA in the course of CKD focuses on restoring the body’s acid–base balance. The goal of MA treatment in patients with CKD is to achieve a bicarbonate concentration in the venous blood plasma equal to or greater than 22 mEq/L [142]. In observational studies, it is suggested that aiming for a bicarbonate serum level close to 28 mEq/L may improve clinical outcomes, but values above 26 mEq/L may be associated with the risk of heart failure and mortality [25,143,144].

In the treatment of MA in patients with CKD, the primary medication is sodium bicarbonate [12,30]. A new drug used in the treatment of acid–base balance disorders in patients with chronic kidney disease is an oral, non-absorbable polymer, veverimer, whose efficacy has been confirmed in clinical trials [145]. Additionally, in selected patients with concomitant volume overload, the use of furosemide may aid in acidosis correction [146]. As a loop diuretic, furosemide induces natriuresis and chloride loss, promoting contraction alkalosis and relative bicarbonate retention, thus contributing to an alkaline shift in acid–base balance [147].

### 12.1. Sodium Bicarbonate

Sodium bicarbonate acts as a buffer that neutralizes excess H^+^ ions, leading to an increase in blood pH and a reduction in MA [148,149,150,151]. Therapy using it may slow the progression of CKD, as confirmed by studies indicating an improvement in the estimated glomerular filtration rate (eGFR) and a reduction in the risk of developing end-stage CKD [151,152,153]. In one study, it was shown that patients treated with this preparation had a lower risk of doubling their creatinine levels and a reduced need for dialysis therapy [154]. However, not all studies have shown significant benefits in kidney function [155]. Sodium bicarbonate may affect the renin–angiotensin system, which is important in the context of blood pressure regulation and kidney function, although not all studies confirm significant changes in the markers of this system [149]. The use of sodium bicarbonate may lead to increased sodium retention and blood pressure, especially in patients who do not adhere to a strict low-sodium diet [151,156,157]. Studies have shown that the preparation does not significantly affect systolic blood pressure if patients are subjected to strict dietary control [157]. Sodium bicarbonate acts as an alkalizing agent, leading to an increase in urine pH. Higher urine pH inhibits the reabsorption of citrate in the renal tubules, resulting in increased citrate excretion in the urine. Citrate binds calcium, preventing the formation of calcium oxalate crystals, so its higher concentration in urine protects against kidney stones. In patients, such supplementation reduces the incidence of kidney stones by 60–80% [158,159,160]. This preparation does not affect the level of uric acid, which is important because its elevated concentration often occurs in patients with CKD [161]. Although this drug increases bicarbonate levels in the serum, no significant differences in muscle function or bone mineral density were observed compared to placebo [162]. During sodium bicarbonate therapy, various side effects such as hypokalemia and hypocalcemia may occur, which in turn can lead to QTc interval prolongation and increase the risk of torsades de pointes, especially in cases of poisoning with drugs acting on sodium and potassium channels such as hydroxychloroquine [163], diphenhydramine [164], and flecainide [163]. In studies involving kidney transplant recipients, sodium bicarbonate therapy was considered safe, not leading to significant changes in transplant function or deterioration of hemodynamic parameters [157,165].

### 12.2. Veverimer

Veverimer acts as a high-capacity and selective polymer that binds and removes HCl from the gastrointestinal tract [33]. After binding with HCl, veverimer increases the excretion of chlorides in the stool, leading to an increase in serum bicarbonate levels and correction of MA [145,166]. In clinical studies, this drug effectively increased serum bicarbonate levels and improved the physical functions of patients with CKD and MA [167]. Compared to placebo, patients treated with veverimer showed a significant increase in bicarbonate levels and improvement in physical function tests, such as the repeated chair stand test [168]. Veverimer is a non-absorbed polymer, which means it is not absorbed from the gastrointestinal tract and is excreted in the feces. Thanks to this, the risk of drug interactions is limited to the potential impact on the absorption of other oral medications by binding to them in the gastrointestinal tract or increasing stomach pH [58,169]. Clinical studies have shown that the given preparation is safe and well-tolerated, and its use is not associated with clinically significant drug interactions [169]. Although it has been shown that it can bind to some negatively charged drugs, such as furosemide or aspirin, these interactions are minimized under physiological conditions with normal chloride concentrations [170]. Additionally, veverimer may temporarily raise gastric pH, but this effect is short-lived and does not cause significant changes in the bioavailability of pH-dependent drugs [169]. In clinical trials, the drug did not show significant adverse effects, and the incidence of headaches and gastrointestinal disorders, such as diarrhea and bloating, was comparable to that observed in the placebo group [171,172]. Veverimer is a promising alternative to sodium bicarbonate in the treatment of MA in patients with CKD, especially for those who need to avoid additional sodium intake.

In the VALOR-CKD study, which included patients with CKD (stage G3) and macroalbuminuria, the impact of veverimer on kidney function was evaluated. Although it did not slow the progression of kidney disease, it was noted that it reduced AG in patients after 5, 12, and 52 weeks of treatment. This may suggest that veverimer affects acid–base balance, potentially improving kidney function. However, the authors of the study note that the small difference in bicarbonate levels between the research groups could have influenced the study’s outcome [173].

Both drugs effectively increase bicarbonate levels, but veverimer offers the additional benefit of no sodium supply, which may be significant for patients with hypertension or other conditions related to sodium retention. Further studies are needed to fully understand the long-term benefits and potential risks associated with its use. [168,171].

## 13. Conclusions

MA is an acid–base balance disorder, and its occurrence is dependent on the stage of chronic kidney disease. It is associated with negative consequences, which can include the development of an inflammatory process, the risk of osteoporosis, anemia, or cardiovascular events. Additionally, it accelerates the progression of CKD. The basis is the treatment of patients, in which the main applications are bicarbonates and veverimer, as well as a diet rich in vegetables and fruits. The results of previous studies confirm that treating metabolic acidosis allows for the preservation of kidney function; however, there is still insufficient evidence that its correction benefits patients. This requires conducting additional clinical studies.

## Figures and Tables

**Figure 1 diagnostics-15-02052-f001:**
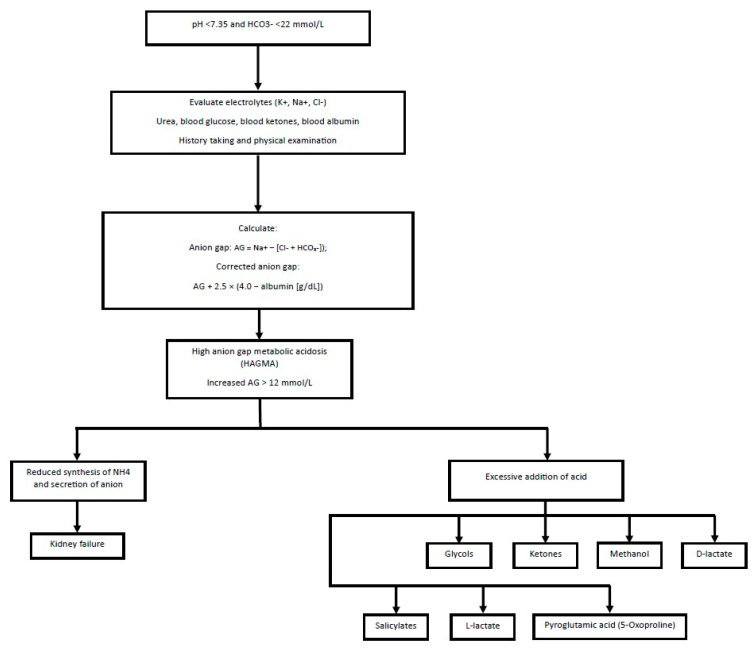
Diagnostic scheme for metabolic acidosis, including HAGMA. Abbreviations: AG, anion gap; Cl^−^, chloride; HCO_3_^−^, bicarbonate; H^+^, hydron; K^+^, potassium; Na^+^, sodium [28,29].

**Figure 2 diagnostics-15-02052-f002:**
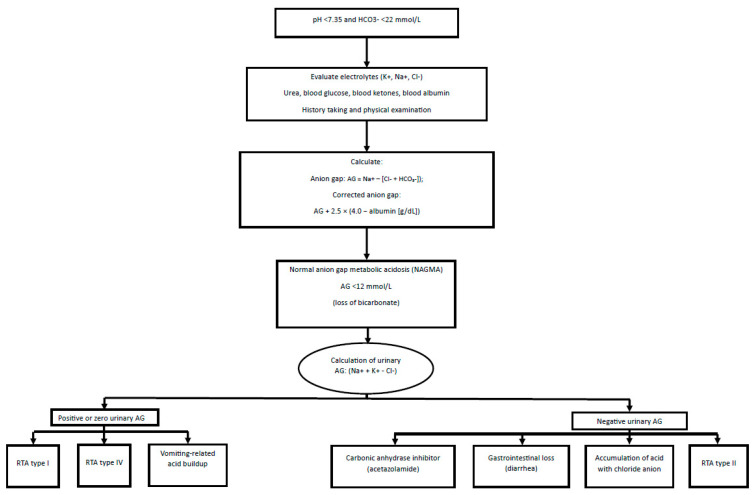
Diagnostic scheme for metabolic acidosis, including NAGMA. Abbreviations: AG, anion gap; Cl^−^, chloride; HCO_3_^−^, bicarbonate; H^+^, hydron; K^+^, potassium; Na^+^, sodium; RTA, renal tubular acidosis [30,31].

**Figure 3 diagnostics-15-02052-f003:**
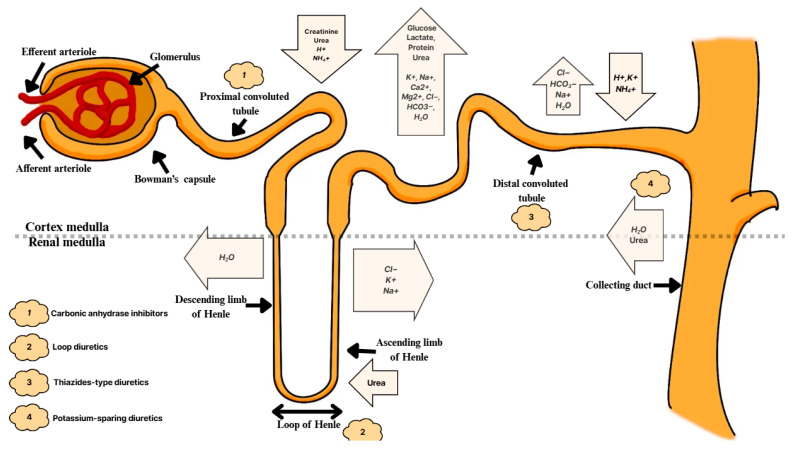
Diagram of the nephron. Points of uptake and reabsorption of ions and chemical compounds. Points of drug uptake. Abbreviations: Cl^−^, chloride; HCO_3_^−^, bicarbonate; H^+^, hydron; K^+^, potassium; Na^+^, sodium; NH_4_^+^, ammonium [3,10,12,54].

**Table 1 diagnostics-15-02052-t001:** Values of clinical parameters in patients with metabolic acidosis, along with the applicable norms.

Parameter	Normal Value	Value in Metabolic Acidosis	Notes
pH	7.35–7.45	<7.35	Diagnosis and monitoring of acid–base disorders
pCO_2_	35–40 mmHg	<35 mmHg	The decline in pCO_2_ is due to compensatory hyperventilation
HCO_3_^−^	21–27 mmHg	<22 mmol/L	A low level indicates a loss of base or an excess of acid
BE	−2.3 do +2.3 mEg/L	<−2 mEg/L	
AG	4 to 12 mmol/L	>12 mmol/L for HAGMA (N) AGMA	It is used to differentiate types of acidosis: elevated AG (e.g., lactic acidosis), normal AG (diarrhea)
Cl^−^	96–108 mEq/L	↑/N	An increase in chlorides indicates hyperchloremic acidosis (AG within normal range)
AGcorr	8–12 mmol/L		AGcorr. is an AG corrected to albuminemia. Hypoalbuminemia masks a high anion gap

Abbreviations: AG, anion gap; AGcorr, corrected anion gap; BE, base excess; Cl^−^, chloride; HCO_3_^−^, bicarbonate; pCO_2_, carbon dioxide partial pressure, ↑, higher than normal.

**Table 2 diagnostics-15-02052-t002:** Diagnosis of metabolic acidosis based on blood gasometry.

	pH	pCO_2_	HCO_3_^−^
Uneven			
	↓	N	↓


Partially balanced			
	↓	↓	↓


Fully balanced			
	N	↓	↓



Abbreviations: HCO_3_^−^, bicarbonate; pCO_2_, carbon dioxide partial pressure; ↓, lower than normal.

**Table 3 diagnostics-15-02052-t003:** Drug-induced renal tubular acidosis.

Drug	Mechanism
Proximal RTA	
Carbonic anhydrase inhibitors (acetazolamide, dorzolamide, and methazolamide)	Causes isolated impairment of bicarbonate reabsorption in the proximal tubule, leading to proximal RTA. This effect is mainly due to the inhibition of the CA IV isoenzyme, without affecting the reabsorption of other dissolved substances. As a result, there is a loss of HCO_3_^−^ in the urine, and the development of hyperchloremic metabolic acidosis with a normal anion gap.
Cisplatin	A direct toxic effect on the proximal convoluted tubule’s amino acid transporter results in toxicity, which causes cell death and renal Fanconi syndrome, impaired resorption of HCO_3_^−^.
Tenofovir	Builds up in the renal proximal tubule cells, where the mechanism is mitochondrial toxicity. Fanconi syndrome results from this, which impairs the reabsorption of bicarbonate, phosphate, glucose, and amino acids.
Ifosfamide	An alkylating agent is used to treat a variety of tumors, including testicular cancer, soft tissue sarcomas, and bone sarcomas. The exact process by which ifosfamide damages renal tubules is unknown.
Distal RTA	
Amphotericin B	Causes back-diffusion of released H^+^ ions and K^+^ squandering, which in turn causes RTA type 1 by increasing membrane permeability in the collecting duct
Foscarnet	It is believed that the mechanism involves mitochondrial malfunction that damages the cells of the renal tubules
Lithium	Lithium medication is thought to cause distal renal tubular acidosis by permitting excessive acid back-diffusion.

**Table 4 diagnostics-15-02052-t004:** Causes leading to the development of HAGMA.

List	Parameter	Development Mechanism HAGMA
G	Glycols	Ethylene and propylene, propylene glycol used as a solvent, for example, in lorazepam or phenobarbital, are metabolized to D-lactate and L-lactate.
O	Pyroglutamic acid (5-Oxoproline)	The mechanism is based on the disruption of the gamma-glutamyl cycle, leading to the production of pyroglutamic acid. The cause is a deficiency of glutathione. The accumulation of 5-oxoprolin occurs especially in malnourished individuals with CKD, liver failure, and those chronically using paracetamol.
L	L-lactate	It occurs with excessive lactate production and impaired hepatic clearance. In conditions of hypoxia, tissue ischemia, and sepsis, glucose is converted to lactic acid by the enzyme lactate dehydrogenase.
D	D-lactate	It can develop in individuals with short bowel syndrome or after bowel resections and bacterial overgrowth in the colon. Among this group of people, undigested starch and glucose are fermented by bacteria in the colon into organic acids, including D-lactate, which is poorly metabolized by humans.
M	Methanol	It is converted in the liver by alcohol dehydrogenase to formic aldehyde, and then by aldehyde dehydrogenase to formic acid, which accumulates in the blood and inhibits mitochondrial cytochrome oxidase, leading to cellular hypoxia.
A	Aspirin	It undergoes hydrolysis to salicylic acid. Salicylates inhibit oxidative phosphorylation in the mitochondria, resulting in increased production of lactic acid and exacerbation of acidosis. They also stimulate lipolysis and ketogenesis, increasing the concentration of ketones.
R	Kidney failure	Acidosis develops due to the retention of organic and inorganic acids. In kidney failure, the accumulation of anions, such as sulfonic acid derivatives (indoxyl, p-cresol), is observed, leading to the accumulation of phosphoric and sulfuric acids.
K	Ketones	Ketone bodies (mainly acetone, acetoacetate, β-hydroxybutyrate) are produced in the liver as a result of ketogenesis—a process that mainly occurs in situations of glucose or insulin availability deficiency. These situations include fasting, untreated type 1 diabetes, or alcoholism.

## Data Availability

Not applicable.

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
