# Peer review of "Metabolic Acidosis in Patients with Chronic Kidney Disease: Diagnosis, Pathogenesis, and Treatment—A Narrative Review"

_diagnostics, 2025, doi:10.3390/diagnostics15162052_

Round 1

Reviewer 1 Report

Comments and Suggestions for Authors

Dear Authors,

Thank you for submitting your manuscript, “Metabolic acidosis in patients with chronic kidney disease: diagnosis, pathogenesis, and treatment – a narrative review.” This review addresses a clinically important topic, offering an updated and comprehensive perspective on metabolic acidosis in CKD, with special emphasis on KDIGO 2024 guidelines, dietary interventions, and emerging therapies like veverimer. Overall, the manuscript is well-written, thoroughly researched, and of significant interest to clinicians and researchers.

I believe the manuscript can be accepted after major revisions. My detailed comments are below:

  1. The manuscript would benefit from a dedicated discussion of medication-induced acidosis, especially focusing on metformin and SGLT2 inhibitors. These agents are widely used in CKD patients and are clinically relevant due to their association with lactic acidosis and euglycemic ketoacidosis.
  2. A focused summary or table of drugs known to cause renal tubular acidosis (RTA) or metabolic acidosis—such as ifosfamide, tenofovir, and acetazolamide—would be valuable for readers and improve the practical utility of the review.
  3. The addition of a diagnostic and management algorithm for metabolic acidosis in CKD would significantly enhance the manuscript’s clinical relevance. Such an algorithm could guide clinicians through stepwise evaluation and treatment strategies.
  4. Including an annotated nephron diagram and a visual infographic summarizing acid-base regulation and treatment options would make the content more engaging and easier to understand.
  5. The review does not address metabolic acidosis in renal transplant recipients. A brief discussion, particularly of tacrolimus-induced RTA, would round out the scope of the review.
  6. The link between metabolic acidosis and kidney stone formation is well established (e.g., via hypocitraturia, low urine pH, and calcium release from bone) but is not mentioned. Including this would add an important clinical dimension.
  7. The alkalinizing effect of furosemide, which can aid in correcting acidosis in volume-overloaded CKD patients, is not discussed. A short note on this point would enhance the section on management.

Reviewer 2 Report

Comments and Suggestions for Authors

This narrative review provides a comprehensive overview of MA in patients with CKD, addressing its definition, pathogenesis, clinical consequences, and treatment approaches. The manuscript is clearly structured and clinically relevant. However, the current version requires major revision to enhance its scientific depth and better align with contemporary clinical evidence and guideline recommendations. Several sections would benefit from more detailed mechanistic discussion and updated clinical perspectives.

  1. Section 2: Metabolic acidosis is not merely a complication of CKD but may also act as a contributing factor to disease progression. The current manuscript could be strengthened by emphasizing this bidirectional relationship. Highlighting this concept would underscore the clinical importance of early recognition and management of MA in slowing CKD progression.
  2. Section 4: The differences in acid–base homeostasis between PD and HD warrant further explanation. Specifically, a mechanistic discussion of how PD solutions—particularly bicarbonate- versus lactate-buffered dialysates—affect systemic acid–base balance would provide valuable clinical insights. This would also support the choice of dialysis modality in managing MA.
  3. Section 6: Patients with CKD are more susceptible to type B lactic acidosis, especially metformin-associated lactic acidosis. The review would benefit from discussing the risk profile and epidemiological patterns of MALA in the context of impaired renal function, which is essential for risk mitigation and clinical decision-making.
  4. Section 7: The discussion on metabolic acidosis and CKD progression should incorporate molecular-level mechanisms of renal ammoniagenesis. Including key enzymes and transporters such as PEPCK, GLS1, and NHE3 would enhance the biological plausibility of how acid-base imbalance drives nephron injury. Additionally, it would be valuable to reference the KDIGO 2020 guidelines, which address the potential benefits of alkali therapy in attenuating CKD progression.

Round 2

Reviewer 1 Report

Comments and Suggestions for Authors

I acknowledge that the authors have thoroughly addressed all my comments and suggestions in the revised manuscript. The responses are satisfactory, and the necessary revisions have been implemented appropriately. Therefore, my final decision is to accept the manuscript for publication.

Reviewer 2 Report

Comments and Suggestions for Authors

Thank you for the revised manuscript. The authors have addressed all previous comments adequately, and the current version has been thoroughly improved. I find the manuscript acceptable in its present form and recommend it for publication.